# OpenReview forum: "Influence Functions in Deep Learning Are Fragile"
_ICLR.cc/2021/Conference — ICLR 2021 Poster_

### Official Review · AnonReviewer2 · 2020-10-27
**influence functions are indeed fragile, but how significant are these results?**

**Rating:** 6
**Confidence:** 3

**Review:**

In this work, the authors perform an empirical analysis of influence functions in deep learning. Influence functions allow one to estimate the effect of weighting a training point at test time (for instance, dropping a training point from the training set with a weight of $\frac{-1}{n}$ under assumptions about the convexity of the loss function and the models Hessian $H_{\theta}$. The authors note that successful applications of influence functions are rare in the literature, despite their potential applications. The authors investigate the effects of model width, depth, regularization schedule, Hessian approximation techniques, and dataset size on the accuracy of influence functions. In general, they find that each of these can cause issues with influence function calculation. As model width and depth increases, the Spearman correlation decreases between the calculated influence functions and the ground truth obtained by retraining. They also find that the top eigenvalue of $H_\theta$ increases quite dramatically as depth increases. Weight regularization was found to be important for successful influence functions calculation, while factors such as the huge size of the ImageNet dataset proved prohibitive at this point.

Strengths:
* The paper has excellent presentation. The authors establish the background and relevant literature succinctly but with proper depth
* I do think that the authors demonstrated that most of the factors they consider do contribute to the difficulty of accurately computing influence functions, but see (weaknesses)


Weaknesses:
* The authors use the term "significant" or "significantly" 10+ times throughout the paper but I think this word should generally be reserved for when a test of statistical significance is performed. Consider using another word such as "substantial"
* Why are there no error bars on Fig 1c-d, Fig 2, and Fig 3? Could you consider rerun your experiments over many different initializations of the model? Since you mainly analyzed results on MNIST and CIFAR-10 that should be feasible.
*Fig 1c and 2c seem particularly stochastic and could benefit from multiple initializations to confirm or deny the trends that are claimed.

Overall, I think this is a fairly strong work that performs an important empirical analysis but I have concerns about the lack of repeated experiments and seeing whether these trends reported hold up over many initializations. I rate it a 6 on this basis.

---

> ### Author Response · Authors · 2020-11-20
> **Responses to your questions**
>
> We thank the reviewer for the insightful comments. Below we summarize our responses:
>
> - “The authors use the term "significant" or "significantly" 10+ times throughout the paper but I think this word should generally be reserved for when a test of statistical significance is performed.”:  Thanks for the suggestion. To avoid confusion, we have replaced the word ‘significantly’ in the updated version of the paper with suitable alternatives.
>
> - “Why are there no error bars on Fig 1c-d, Fig 2, and Fig 3 / Fig 1c and 2c seem particularly stochastic ”: We have repeated our experiments for Fig. 1 and Fig. 2 with different runs/initialisations and updated the relevant figures in the paper with error bars. Across the different experiments, we find that the error bars are small in general. We have also conducted further experiments on small MNIST (in a similar setting as Fig. 3) with different runs and reported the correlation estimates with error bars in Fig. 11- Appendix. As before, we find the error bars to be relatively small.

---

### Official Review · AnonReviewer4 · 2020-10-28
**Review on "Influence functions in deep learning are fragile"**

**Rating:** 6
**Confidence:** 4

**Review:**

##########################################################################

Summary:
The paper provides an extensive experimental study on using influence functions in deep neural networks. The authors suggest that the estimation via influence functions can be quite fragile depending on various architectures of neural networks. To point out such information, approximation with inverse-Hessian Vector product techniques might be incorrect, which leads to low quality influence estimates. They show several meaningful findings which address the true nature of using influence functions in deep neural networks. One of their findings through various experimental study is that the network depth and width strongly affect influence estimates.

##########################################################################

Reasons for score:
The introduced through experimental settings seem to be highly interesting and reasonable. I like the idea of addressing weakness of influence functions in deep neural network and the authors’ findings are interesting. I believe it will be better if the authors perform the ablation studies with different parameters and more datasets to generalize their findings.

##########################################################################

Pros:
1. The paper address the important issue of influence functions, which can affect the field of model interpretability.
2. The introduced experimental settings seem to be reasonable to empirically show that using influence functions in deep neural networks can be very sensitive depending on the architectures of neural networks, the use of weight decay, and the choice of hyper-parameters.
3. The experiments are well designed with multiple architectures of neural networks and detasets.

In the case of non-convex loss functions, the assumption of fist-order influence functions is not generally true. They empirically found that the Taylor’s gap is strongly affected by common hyper-parameters for deep networks.

##########################################################################

Cons:
More ablation studies with different set of hyper parameters and multiple datasets can improve the quality of this paper.

##########################################################################

Questions during rebuttal period:
In the case study using a CNN architecture on the small MNIST, the authors need to perform the experiments with different number of selected training samples with both the highest and the lowest influence scores to amplify their point.

#########################################################################

---

> ### Author Response · Authors · 2020-11-20
> **Responses to your questions**
>
> Thank you for your detailed comments and positive feedback on the relevance of our work. Below we summarize our responses:
>
> - “More ablation studies with different set of hyper parameters and multiple datasets can improve the quality of this paper”:  Thank you for the suggestion to include experimental results on more datasets to strengthen our paper. We further investigated the quality of influence estimates for CIFAR-100. This is a more challenging dataset than CIFAR-10, but less complex than ImageNet-1000, both for which we have presented results in the paper. With ResNet-18, when trained with a weight-decay factor of 0.0005, across multiple test-points and initialisations, we find that the influence estimates have poor quality. This is similar to our observations with ImageNet, thus holding up the fragile notion of influence functions especially for large models and datasets. In the case of CIFAR-100, for certain test-points, we notice inconsistencies in the semantic similarity of the top influential points to the given test-point (similar to our observations with CIFAR-10 and ImageNet experimental results). We have included these results along with a discussion in the updated version of the paper (See Fig. 4, Sec 5.3 ; Section 11 - Appendix).
>
> - “In the case study using a CNN architecture on the small MNIST, the authors need to perform the experiments with different number of selected training samples with both the highest and the lowest influence scores to amplify their point”:  Upon your suggestion, we have added new experiments for the CNN architecture trained on small MNIST with different training samples (with the highest and lowest influence scores respectively). We compute the correlation estimates with the top and the bottom [10,20,40,60,80,100,120] influential training samples with three different initialisations. We find that the influence estimates for training samples with the highest influence scores are good (i.e. high correlations with the respective ground-truths). However influence estimates for the training samples with the lowest influential scores are poor (i.e. low correlation with the respective ground-truths).  For the lowest influential training samples, we observe that the influence estimates suffer from large approximation errors and are of the order of 1e-7, in contrast to the ground-truth estimates which are of the order of 1e-3. A detailed result and discussion has been added in the updated paper (See Sec 9 - Appendix).

---

### Official Review · AnonReviewer1 · 2020-10-28
**Summary: The paper provides an empirical analysis of using influence functions in the context of neural network architectures. A detailed empirical study on shallow nets is provided to support the findings.**

**Rating:** 6
**Confidence:** 5

**Review:**

Strength:

+ The paper provides an interesting application of Influence functions that were introduced in Koh et al. for studying data poisoning attacks. The main idea of the paper given in Section 3 is that an approximation of the influence or impact of a training sample on the test set can be obtained using second order Taylor's series expansion at the optimal model parameter -- a bilinear form where the matrix of the form is given by the hessian inverse and the vectors are gradients evaluated using training and test sample. However, since this approximation only holds when the Hessian positive definite, the paper identifies certain settings in which the influence functions can be estimated to reasonable approximation.

Weaknesses:

- The "retraining" phase to get the ground truth seems like an interesting idea, which in my understanding has not been looked at. However, this retraining phase has been directly borrowed from Koh et al., and in fact, Koh et al propose the method also as an approximation. The paper fails to clarify why such an approximation will provide any insights on the performance of neural networks. Including a summary, and intuition of the approach will be helpful to the reader.
- Most of the neural networks used have a relu activations (with linear layers), making them a piecewise linear function with respect to the parameters, more importantly, nonsmooth. In fact, in it possible to count the number of linear regions, see (http://proceedings.mlr.press/v80/serra18b.html). In this light, it is unclear what the main message of the paper from an intuitive perspective is. Moreover, the paper misses an important line of work in statistics called Experimental Design in which the goal is to approximate a dataset using a subset with respect to various statistical criterion on the estimators, see https://www.jstor.org/stable/2290334?seq=1.
- Related to the previous point: It is unclear what the take away is from the experimental results in the paper. Some of the experiments are performed using very shallow networks (width of 5, depth of 1 and relu activations) whereas others have a few more layers. However, two test points are used in all the experiments. Why? All of the datasets used in the experiments have thousands of points in each class, and it is also possible to augment them by simple rotations, translations etc.. The paper will benefit by adding robust estimates of Pearson and Spearman using a fraction of test samples around the highest and 50% test points.
- Finally, none of the results are provided with confidence intervals over runs. Are the experiments averaged over a few runs? If so, I suggest you include them in the figures to provide an accurate summary of the experiments.

After response: Thanks for the clarifications especially the piecewise linearity of deep network as a function of input and not as parameters (like I incorrectly suggested in my review). I think the empirical results have some merit, but are preliminary and I fail to see the bigger picture. For example, more carefully designed ablation studies are needed to determine the practical utility i.e., when the approximations suggested are reliable and when they are not.

---

> ### Author Response · Authors · 2020-11-20
> **Responses to your questions**
>
> We thank the reviewer for detailed comments on our work. Below we summarize our responses:
>
> - “The ‘retraining’ phase to get the ground truth seems like an interesting idea... The paper fails to clarify why such an approximation will provide any insights on the performance of neural networks. Including a summary, and intuition of the approach will be helpful to the reader”:   To evaluate the ground-truth influence estimates, re-training from optimal model parameters is a proposal which we borrow from [1]. Note that training from scratch to obtain the ground-truths even for small models and datasets is very computationally expensive, that is why such an approximation is proposed in the literature. To further validate the approach followed by [1], we carried out two new experiments:
>
>    (a) For a 4-layered shallow network trained on Iris, we observe that both training from scratch and re-training from optimal model parameters lead to similar correlation estimates across different initialisations (See Fig. 5(b)).  We also compute the norm of the difference in parameters obtained by training from scratch vs. re-training from optimal model parameters as suggested by [1].  For the same network trained on Iris, we observe that the norm of the difference in parameters is relatively small (in the order of $e^{-7}$) across different training samples (See Fig. 5(a)).
>
>    (b) Similarly for a CNN architecture (with depth 6) trained on MNIST, the norm of the difference in parameters is also small and is of the order of $e^{-6}$ (See Fig. 14 - Appendix).  These results highlight that re-training from optimal parameters (although an approximation) is indeed close to training from scratch but saves substantially in terms of running time.
>
>   The impact of a training sample on the model is often understood with an answer to the counterfactual question: “How would the model change if a certain training sample was excluded?”. The answer to this counterfactual question leads to a fine-grained understanding of model behaviour and its performance [1,2,3]. Training a model from scratch (although prohibitively expensive) by excluding a particular training sample is one way to answer this counterfactual. In our new experiments, we validate that re-training from optimal parameters (as proposed in [1]) closely approximates training the model from scratch, thus answering the same counterfactual question which provides insights into the model performance.
>
>
> - “Most of the neural networks used have a relu activations (with linear layers), making them a piecewise linear function with respect to the parameters, more importantly, nonsmooth. In fact, in it possible to count the number of linear regions, see (http://proceedings.mlr.press/v80/serra18b.html). In this light, it is unclear what the main message of the paper from an intuitive perspective is” :   We agree that global influence analysis of the ReLU networks has an additional challenge since there are measure zero subsets where the function is non differentiable. However, as in the original influence function paper [1], this is often ignored in experiments since the probability of hitting these non differentiable regions is zero (note that ReLU networks are piecewise linear in the input space, they are not in the parameter space). Moreover, although ReLU is a non-smooth activation function, we still observe that in practice for certain types of networks (See Table 1 and Fig. 1-Appendix), the influence estimates are accurate. However through an extensive empirical study over progressively growing networks and datasets, we find that depth, width and overparameterization are the main controlling factors for the quality of influence estimates. We have also conducted experiments using smooth activations which show similar trends to what we observe on ReLU networks (see Fig. 3 where softplus is used and Fig. 2 -Appendix where tanh is used). While it is interesting to understand the impact of the number of linear regions in ReLU networks on the influence estimates through a careful ablation study, we defer it for future work. We have added this discussion and cited the reference you mentioned about the analysis of ReLU networks.
>
>
> [1]. P. W. Koh and P. Liang. Understanding black-box predictions via influence functions. In International Conference on Machine Learning (ICML), 2017.
>
> [2]. Ryan Giordano, Will Stephenson, Runjing Liu, Michael I. Jordan, and Tamara Broderick. A swiss army infinitesimal jackknife. In AISTATS, 2018
>
> [3].  Pang Wei Koh, Kai-Siang Ang, Hubert H. K. Teo, and Percy Liang. On the accuracy of influence functions for measuring group effects. CoRR, abs/1905.13289, 2019b. URL http://arxiv.org/abs/1905.13289.
>
> [4]. Thiago Serra, Christian Tjandraatmadja, and Srikumar Ramalingam. Bounding and counting linear regions of deep neural networks. arXiv preprint arXiv:1711.02114, 2017.

---

> > ### Author Response · Authors · 2020-11-20
> > **Further responses to your comments**
> >
> > - “ the paper misses an important line of work in statistics called Experimental Design in which the goal is to approximate a dataset using a subset with respect to various statistical criterion on the estimators”:
> > Thank you for pointing us to this reference. While some of the ideas presented in the paper could be potentially adapted in ways similar to influence functions for understanding model behaviour, it is outside the scope of our current work. However the core idea of approximating a dataset is closely related to influence functions and therefore we have cited this paper in Sec. 2.
> >
> >
> >
> > - “Some of the experiments are performed using very shallow networks (width of 5, depth of 1 and relu activations) whereas others have a few more layers.”:  We would like to point out that in addition to the analysis of influence estimates with shallow networks, we have presented extensive experimental results with deep networks (e.g. ResNet-50) and complex datasets (to the scale of ImageNet). The reason we started explaining our findings in shallow networks first was that for shallow networks the exact Hessian can be computed and thus can be used to understand the role of approximate Hessian-vector product estimators in influence estimations (See Sec 5.3,  5.4, Table1, Fig. 4 ).
> >
> >
> > - “The paper will benefit by adding robust estimates of Pearson and Spearman using a fraction of test samples around the highest and 50% test points”: Influence computation for multiple test-points is expensive, therefore we restrict our analysis to two test samples for large models and datasets.  Upon your suggestion, we have added new experiments with correlation estimates evaluated on a set of test-points with the highest test-loss, $75^{th}$ percentile of the test-loss, $50^{th}$ percentile of the test-loss and the lowest test-loss (See Sec 13 - Appendix). We find that the correlations computed with multiple test-points are relatively similar to the ones computed with a single test-point.
> >
> > - “Finally, none of the results are provided with confidence intervals over runs”:  Thank you for pointing this out. We have repeated our experiments with different runs/initialisations and updated the relevant figures with error bars in the main paper (See updated Fig.1 and Fig. 2).   We find that in general the error bars are relatively small across the different experiments presented in the paper. Additionally we have reported the new experiments requested in the rebuttal with error bars (See Fig. 4, Fig. 5, Fig. 3 - Appendix, Fig. 11-Appendix, Fig. 13-Appendix)

---

### Official Review · AnonReviewer3 · 2020-10-31
**Strong Empirical Investigation of Influence Functions for DL**

**Rating:** 7
**Confidence:** 3

**Review:**

Summary: The paper does a thorough investigation of influence functions applied to neural networks. Specifically, the authors find that deeper networks result in erroneous influence estimates and weight-decay regularization can help lower the gap in the first-order Taylor expansion of the influence estimates. They also study the empirical effects of network depth and width on the quality of influence estimates. They conclude that influence estimates for smaller networks are quite accurate but are erroneous for larger networks.

Strengths
- Thank you for the specificity in the empirical setup of all the experiments. This should greatly help with the reproducibility of your results.
- The paper clearly defines the influence function set up and notes the shortcomings in estimating it. The authors do a thorough job of discussing how each hyperparameter of network training affects the influence estimates.

Weaknesses
- I would have loved to see some more discussion on why weight decay led to improved influence estimates. Are there some properties of the weight-decayed Hessian that make it more reliable?
- The fragility of influence on large networks has been folk knowledge in the community, and the authors do a good job of codifying this; however, I feel providing theoretical intuitions for the observed phenomena would strengthen the paper's findings. I.e. does overparametrization always lead to imprecise influence estimates?

Question
- Do you intend to release the code for all experiments? For such an empirical paper, this would be strongly encouraged and beneficial for the community.
- Can you provide intuition or details for how to scale influence to larger networks? Is it possible to develop some theory to determine when influence estimates will err (perhaps in terms of the number of parameters, number of training points, etc.)?
- Did you ever run group influence estimation? You mention it in 5.4, but would be nice to see if the same results hold for the influence of a set of points.

*** Post Rebuttal: Thank you so much for your response and for the codebase link. Your additional experiments are convincing for group influence and overparametrization.  I sustain my recommendation to accept this paper :) ***

---

> ### Author Response · Authors · 2020-11-20
> **Responses to your comments**
>
> Thank you for the detailed and insightful feedback on our work. Below we summarize our responses:
>
> - “I would have loved to see some more discussion on why weight decay led to improved influence estimates”: In our experiments, we observe that for deeper networks where the quality of influence estimation is inferior, a high-value of weight-decay helps. In those networks, a relatively high-value of weight-decay decreases the curvature values of the loss function (therefore reduces the Taylor’s approximation gap in influence estimation) and improves the quality of influence estimates. We have updated the paper adding this new experiment (on the Iris dataset) and this additional discussion (See Sec 3-Appendix).
>
> - “Do you intend to release the code for all experiments”:  Yes! we are working on making our codebase user-friendly and will release the code by the end of the review period.
>
> - “Can you provide intuition or details for how to scale influence to larger networks?”: Improving and scaling influence estimates for large overparameterized networks is an important research direction. In our work, we observe that a high value of loss curvature reduces the quality of influence estimates, especially for deeper networks. While outside the scope of the current paper, we propose to train deep networks with curvature regularization in the parameter space to improve influence estimates in deeper models (by reducing the Taylor’s approximation gap in influence estimation).
>
> - “Did you ever run group influence estimation? ”: Influence estimation for groups, especially in the non-convex regime is indeed a challenging problem. (First-order) influence functions assume that the changes in the model on up-weighting a training example is infinitesimal [1,2]. However, in the case of up-weighting a group (instead of a single sample), the model perturbation is substantially large. Upon your suggestion, we have conducted new experiments on group influence estimation using the method proposed by [2] on Iris, MNIST and CIFAR-100. In our new experiments, we observe that for small networks where the exact Hessian matrix can be computed, group influence is fairly accurate for small group sizes. However, even for small models, group influence estimates are inaccurate for large group sizes where the perturbation to the model is considerably large.  For large overparameterized networks, we notice that the group influence estimations are still erroneous for both small and large groups. We have updated the paper with the additional experiments on group influence (See Sec 12 - Appendix).
>
>
> - “Providing theoretical intuitions for the observed phenomena would strengthen the paper's findings. I.e. does overparameterization always lead to imprecise influence estimates” : We agree that understanding the theoretical underpinnings behind the observations in our paper is an extremely interesting and important research direction. While our paper is empirical, we do provide some intuitions about why influence estimates are erroneous for certain networks (e.g. importance of the Taylor’s approximation gap).  To further understand if overparameterization leads to imprecise influence estimates, we perform an additional experiment. We take a simple one-layered network and increase its width gradually from 8 to 300. We observe that the correlations decrease from 0.8 to 0.18 (Fig. 15 - Appendix), which indeed points to the fact that overparameterization plays an important role in controlling the quality of influence estimates.
>
> [1]. P. W. Koh and P. Liang. Understanding black-box predictions via influence functions. In International Conference on Machine Learning (ICML), 2017.
>
> [2]. P. W. Koh, K. Ang, H. H. K. Teo, and P. Liang. On the accuracy of influence functions for measuring group effects. CoRR, abs/1905.13289, 2019a. URL http: //arxiv.org/abs/1905.13289.

---

### Decision · Program_Chairs · 2021-01-07
**Final Decision**

**Decision:**

Accept (Poster)

**Comment:**

This paper examines under what conditions influence estimation can be applied to deep networks and finds that, among of items, that influence estimates are poorer for deeper architectures, perhaps due to poor inverse Hessian vector approximations for poor for deeper models. The authors provide an extensive experimental evaluation across datasets and architectures, and demonstrates the fragility of influence estimates in a number of conditions. Although the reviewers noted that these issues are now "folk knowledge", there has been less scientific effort in identifying these failures.

Of course, more theoretical understanding would help the community better understand where these fragilities lie, but the experimental evaluation is sufficiently strong to be of broad interest to the community.